# Hierarchical and Interpretable Skill Acquisition in Multi-task Reinforcement Learning

**Tianmin Shu**[*]
University of California, Los Angeles
tianmin.shu@ucla.edu

**Caiming Xiong**[†]**& Richard Socher**
Salesforce Research
{cxiong, rsocher}@salesforce.com

## Abstract

Learning policies for complex tasks that require multiple different skills is a major challenge in reinforcement learning (RL). It is also a requirement for its deployment in real-world scenarios. This paper proposes a novel framework for efficient multi-task reinforcement learning. Our framework trains agents to employ hierarchical policies that decide when to use a previously learned policy and when to learn a new skill. This enables agents to continually acquire new skills during different stages of training. Each learned task corresponds to a human language description. Because agents can only access previously learned skills through these descriptions, the agent can always provide a human-interpretable description of its choices. In order to help the agent learn the complex temporal dependencies necessary for the hierarchical policy, we provide it with a stochastic temporal grammar that modulates when to rely on previously learned skills and when to execute new skills. We validate our approach on Minecraft games designed to explicitly test the ability to reuse previously learned skills while simultaneously learning new skills.

## 1 Introduction

Deep reinforcement learning has demonstrated success in policy search for tasks in domains like game playing (Mnih et al., 2015; Silver et al., 2016; 2017; Kempka et al., 2016; Mirowski et al., 2017) and robotic control (Levine et al., 2016a;b; Pinto & Gupta, 2016). However, it is very difficult to accumulate multiple skills using just one policy network Teh et al. (2017). Knowledge transfer techniques like distillation (Bengio, 2012; Rusu et al., 2016; Parisotto et al., 2016; Teh et al., 2017) have been applied to train a policy network both to learn new skills while preserving previously learned skill as well as to combine single-task policies into a multi-task policy. Existing approaches usually treat all tasks independently. This often prevents full exploration of the underlying relations between different tasks. They also typically assume that all policies share the same state space and action space. This precludes transfer of previously learned simple skills to a new policy defined over a space with differing states or actions.

When humans learn new skills, we often take advantage of our existing skills and build new capacities by composing or combining simpler ones. For instance, learning multi-digit multiplication relies on the knowledge of single-digit multiplication; learning how to properly prepare individual ingredients facilitates cooking dishes based on complex recipes.

Inspired by this observation, we propose a hierarchical policy network which can reuse previously learned skills alongside and as subcomponents of new skills. It achieves this by discovering the underlying relations between skills.

To represent the skills and their relations in an interpretable way, we also encode all tasks using human instructions such as "put down." This allows the agent to communicate its policy and generate plans using human language. Figure 1 illustrates an example: given the instruction "Stack blue," our hierarchical policy learns to compose instructions and take multiple actions through a multi-level hierarchy in order to stack two blue blocks together. Steps from the top-level policy $\pi_3$ (i.e., the red

---

[*]This work was done when the author was an intern at Salesforce Research.
[†]Corresponding author

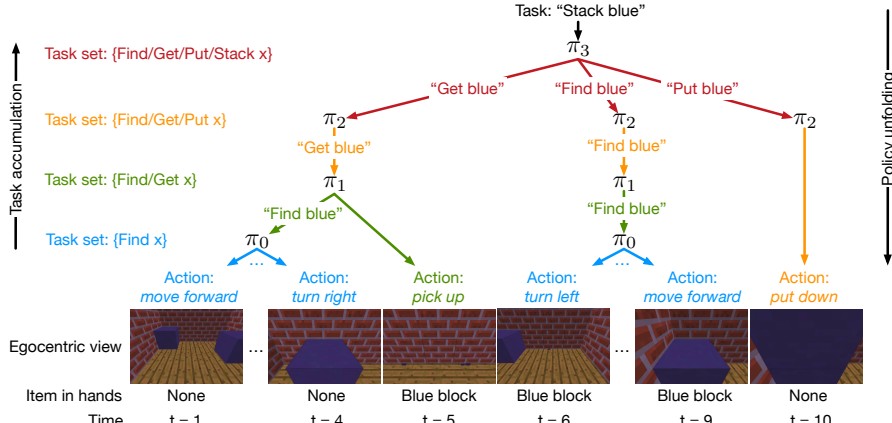

Figure 1: Example of our multi-level hierarchical policy for a given task – stacking two blue blocks. Each arrow represents one step generated by a certain policy and the colors of arrows indicate the source policies. Note that at each step, a policy either utters an instruction for the lower-level policy or directly takes an action.

branches) outline a learned high-level plan – "Get blue → Find blue → Put blue." In addition, from lower level policies, we may also clearly see composed plans for other tasks. Based on policy $\pi_2$, for instance, the task "Get blue" has two steps – "Find blue → action: turn left," whereas "Put blue" can be executed by a single action "put down" according to $\pi_3$. Through this hierarchical model, we may i) accumulate tasks progressively from a terminal policy to a top-level policy and ii) unfold the global policy from top-level to basic actions.

In order to better track temporal relationships between tasks, we train a stochastic temporal grammar (STG) model on the sequence of policy selections (previously learned skill or new skill) for positive episodes. The STG focuses on modeling priorities of tasks: for example, it is necessary to obtain an object before putting it down. Integrating the STG into the hierarchical policy boosts efficiency and accuracy by explicitly modeling such commonsense world knowledge.

We validated our approach by testing it on object manipulation tasks implemented in a Minecraft world. Our experimental results demonstrate that this framework can (i) efficiently learn hierarchical policies and representations for multi-task RL; (ii) learn to utter human instructions to deploy pretrained policies, improve their explainability and reuse skills; and (iii) learn a stochastic temporal grammar via self-supervision to predict future actions.

## 2 Related Work

**Multi-task Reinforcement Learning**. Previous work on multi-task reinforcement learning mainly falls into two families: knowledge transfer through distillation (Rusu et al., 2016; Parisotto et al., 2016; Teh et al., 2017; Tessler et al., 2017) or modular policy design through 2-layer hierarchical policy (Andreas et al., 2017). Our multi-level policy is more similar to the latter approach. The main differences between our model and the one in Andreas et al. (2017) are two-fold: i) we do not assume that a global task can be executed by only performing predefined sub-tasks; ii) in our multi-level policy, global tasks at a lower-level layer may also be used as sub-tasks by global tasks carried out at higher-levels.

**Hierarchical Reinforcement Learning**. Complex policies often require the modeling of longer temporal dependencies than what standard Markov decision processes (MDPs) can capture. To combat this, hierarchical reinforcement learning was introduced to extend MDPs to semi-MDPs (Sutton et al., 1999), where options (or macro actions) are introduced on top of primitive actions to decompose the goal of a task into multiple subgoals. In hierarchical RL, two sets of policies are trained: local policies that map states to primitive actions for achieving subgoals, and a global policy that initiates suitable subgoals in a sequence to achieve the final goal of a task (Bacon & Precup, 2015; Kulkarni et al., 2016; Vezhnevets et al., 2016; Tessler et al., 2017; Andreas et al.,

2017). This two-layer hierarchical policy design significantly improves the ability of discovering complex policies which can not be learned by flat policies. However, it also often makes some strict assumptions that limit its flexibility: i) a task's global policy cannot use a simpler task's policy as part of its base policies; ii) a global policy is assumed to be executable by only using local policies over specific options, e.g., (Kulkarni et al., 2016; Andreas et al., 2017). In this work, we aim to learn a multi-level global policy which does not have these two assumptions. In addition, previous work usually uses a latent variable to represent a task. In our work, we encode a task by a human instruction to learn a task-oriented language grounding as well as to improve the interpretability of plans composed by our hierarchical policies.

**Language grounding via reinforcement learning**. Recently, there has been work on grounding human language in 3D game environments (Hermann et al., 2017; Chaplot et al., 2017) or in text-based games (Narasimhan et al., 2015) via reinforcement learning. In these games agents are instructed to pick up an item described by a sentence. Besides visual grounding, Andreas et al. (2017) grounded instructions (not necessarily using human language) to local policies in hierarchical reinforcement learning. Our approach not only learns the language grounding for both visual knowledge and policies, but is also trained to utter human instructions as an explicit explanation of its decisions to humans. To our knowledge, this is the first model that learns to compose plans for complex tasks based on simpler ones which have human descriptions.

## 3 MODEL

In this section, we discuss our multi-task RL setting, hierarchical policy, stochastic temporal grammar, and how interaction of these components can achieve plan composition.

### 3.1 MULTITASK RL SETTING

Let $\mathcal{G}$ be a task set, where each task $g$ is uniquely described by a human instruction. For simplicity, we assume a two-word tuple template consisting of a skill and an item for such a phrase, i.e., $\langle u_{\text{skill}}, u_{\text{item}} \rangle$. Each tuple describes an object manipulation task. In this paper, we define $g = \langle u_{\text{skill}}, u_{\text{item}} \rangle$ by default, thus tasks and instructions are treated as interchangeable concepts.

For each task, we define a Markov decision process (MDP) represented by states $s \in \mathcal{S}$ and primitive actions $a \in \mathcal{A}$. Rewards are specified for goals of different tasks, thus we use a function $R(s, g)$ to signal the reward when performing any given task $g$.

We assume that as a starting point, we have a terminal policy $\pi_0$ (as shown in Figure 2a) trained for a set of basic tasks (i.e., a terminal task set $\mathcal{G}_0$). The task set is then progressively increased as the agent is instructed to do more tasks by humans at multiple stages, such that $\mathcal{G}_0 \subset \mathcal{G}_1 \subset \cdots \subset \mathcal{G}_K$, which results in life-long learning of polices from $\pi_0$ for $\mathcal{G}_0$ to $\pi_K$ for $\mathcal{G}_K$ as illustrated by the "task accumulation" direction in Figure 1. At stage $k > 0$, $\mathcal{G}_{k-1}$ is defined as the base task set of $\mathcal{G}_k$. The tasks in $\mathcal{G}_{k-1}$ are named as base tasks at this stage and $\pi_{k-1}$ becomes the base policy of $\pi_k$. Here, we utilize weak supervision from humans to define what tasks shall be augmented to the previous task set at each new stage.

### 3.2 HIERARCHICAL POLICY

One of our key ideas is that a new task in current task set $\mathcal{G}_k$ may be decomposed into several simpler subtasks, some of which can be base tasks in $\mathcal{G}_{k-1}$ executable by base policy $\pi_{k-1}$. Therefore, instead of using a flat policy (Figure 2a) as $\pi_0$ that directly maps state and human instruction to a primitive action, we propose a hierarchical design (Figure 2b) with the ability to reuse the base policy (i.e., $\pi_{k-1}$) for performing base tasks as subtasks. Namely, at stage $k$, the global policy $\pi_k$ is defined by a hierarchical policy. This hierarchy consists of four sub-policies: a base policy for executing previously learned tasks, an instruction policy that manages communication between the global policy and the base policy, an augmented flat policy which allows the global policy to directly execute actions, and a switch policy that decides whether the global policy will primarily rely on the base policy or the augmented flat policy.

The base policy is defined to be the global policy at the previous stage $k - 1$. The instruction policy maps state $s$ and task $g \in \mathcal{G}_k$ to a base task $g' \in \mathcal{G}_{k-1}$. The purpose of this policy is to inform base

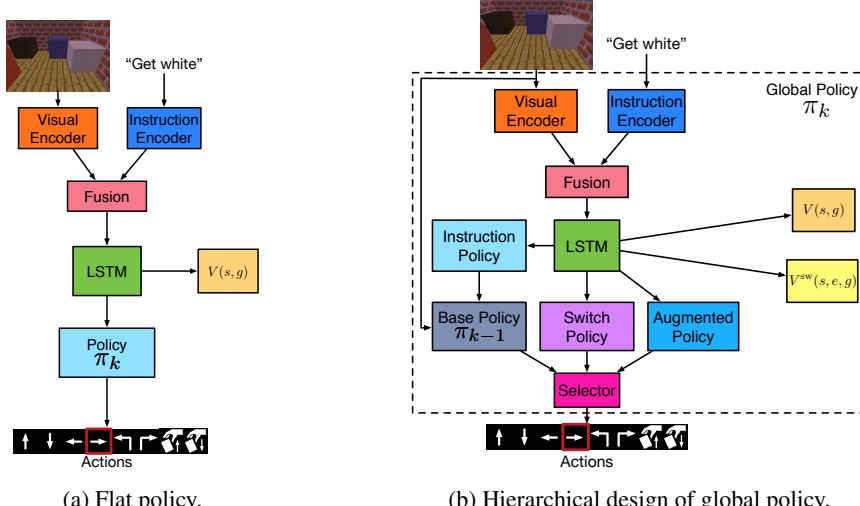

(a) Flat policy.  (b) Hierarchical design of global policy.

Figure 2: Flat and hierarchical policy architectures. $V(s, g)$ and $V^{\text{sw}}(s, e, g)$ are value functions defined in Section 4.1.

policy $\pi_{k-1}$ which base tasks it needs to execute. Since an instruction is represented by two words, we define the instruction policy using two conditionally independent distributions, i.e., $\pi_k^{\text{inst}}(g' = \langle u_{\text{skill}}, u_{\text{item}} \rangle | s, g) = p_k^{\text{skill}}(u_{\text{skill}} | s, g) p_k^{\text{item}}(u_{\text{item}} | s, g)$. An augmented flat policy, $\pi_k^{\text{aug}}(a | s, g)$, maps state $s$ and task $g$ to a primitive action $a$ for ensuring that the global policy is able to perform novel tasks in $\mathcal{G}_k$ that can not be achieved by only reusing the base policy. To determine whether to perform a base task or directly perform a primitive action at each step, the global policy further includes a switch policy, $\pi_k^{\text{sw}}(e | s, g)$, where $e$ is a binary variable indicating the selection of the branches, $\pi_k^{\text{inst}}$ ($e = 0$) or $\pi_k^{\text{aug}}$ ($e = 1$).

Note that the above description of the hierarchical policy does not account for an STG. The instruction policy and switch policy introduced here are simplified from the ones in the full model (see Section 3.3).

At each time step, we first sample $e_t$ from our switch policy $\pi_k^{\text{sw}}$ to decide whether the global policy $\pi_k$ will rely on the base policy $\pi_{k-1}$ or the augmented flat policy $\pi_k^{\text{aug}}$. We also sample a new instruction $g'_t$ from our instruction policy $\pi_k^{\text{inst}}$ in order to sample actions from the base policy. This can be summarized as:

$$e_t \sim \pi_k^{\text{sw}}(e_t | s_t, g), \tag{1}$$

$$g'_t \sim \pi_k^{\text{inst}}(g'_t | s_t, g), \tag{2}$$

and finally

$$a_t \sim \pi_k(a_t | s_t, g) = \pi_{k-1}(a_t | s_t, g'_t)^{(1-e_t)} \pi_k^{\text{aug}}(a_t | s_t, g)^{e_t}, \tag{3}$$

where $\pi_k$ and $\pi_{k-1}$ are the global policies at stage $k$ and $k-1$ respectively. After each step, we will also obtain a reward $r_t = R(s_t, g)$.

### 3.3 STOCHASTIC TEMPORAL GRAMMAR

Different tasks may have temporal relations. For instance, to move an object, one needs to first find and pick up that object. There has been previous research (Si et al., 2011; Pirsiavash & Ramanan, 2014) using stochastic grammar models to capture such temporal relations. Inspired by this, we summarize temporal transitions between various tasks with a stochastic temporal grammar (STG). In our full model, the STG interacts with the hierarchical policy described above through the modified switch policy and instruction policy by using the STG as a prior. This amounts to treating the past history of switches and instructions in positive episodes as a guidance on whether the hierarchical policy should defer to the base policy to execute a specific base task or employ its own augmented flat policy to take a primitive action.

In an episode, the temporal sequence of $e_t$ and $g'_t$, i.e., $\{\langle e_t, g'_t \rangle; t \geq 0\}$, can be seen as a finite state Markov chain (Baum & Petrie, 1966). Note that the state here is referred to the tuple $\langle e_t, g'_t \rangle$, which is not the state of the game $s_t \in \mathcal{S}$ defined in Section 3.1. Consequently, at each level $k > 0$, we may define an STG of a task $g$ by i) transition probabilities, $\rho_k(e_t, g'_t | e_{t-1}, g'_{t-1}, g)$, and ii) the distribution of $\langle e_0, g'_0 \rangle$, $q_k(e_0, g'_0 | g)$, all of which follow categorical distributions.

With the estimated probabilities, we sample $e_t$ and $g'_t$ in an episode at level $k > 0$ w.r.t. to reshaped policies $\pi_k^{\mathrm{sw}'}$ and $\pi_k^{\mathrm{inst}'}$ respectively:

- If $t = 0$,

$$e_0 \sim \pi_k^{\mathrm{sw}'}(e_0|s_t, g) \propto \pi_k^{\mathrm{sw}}(e_0|s_t, g) \sum_{g' \in \mathcal{G}_{k-1}} q_k(e_0, g'|g), \tag{4}$$

$$g'_0 \sim \pi_k^{\mathrm{inst}'}(g'_0|s_t, g) \propto \pi_k^{\mathrm{inst}}(g'_0|s_t, g) q_k(e_0 = 0, g'_0|g); \tag{5}$$

- Otherwise,

$$e_t \sim \pi_k^{\mathrm{sw}'}(e_t|e_{t-1}, g'_{t-1}, s_t, g) \propto \pi_k^{\mathrm{sw}}(e_t|s_t, g) \sum_{g' \in \mathcal{G}_{k-1}} \rho_k(e_t, g'|e_{t-1}, g'_{t-1}, g), \tag{6}$$

$$g'_t \sim \pi_k^{\mathrm{inst}'}(g'_t|e_{t-1}, g'_{t-1}, s_t, g) \propto \pi_k^{\mathrm{inst}}(g'_t|s_t, g) \rho_k(e_t = 0, g'_t|e_{t-1}, g'_{t-1}, g). \tag{7}$$

Note that primitive action sampling is not affected by the STG.

## 3.4 Plan Composition

Combined with our hierarchical policy and STG defined above, we are able to run an episode to compose a plan for a task specified by a human instruction. Algorithm 1 in Appendix A summarized this procedure with respect to the policy and STG at level $k$. Note that to fully utilize the base policy, we assume that once triggered, a base policy will play to the end before the global policy considers the next move.

## 4 Learning

The learning algorithm is outlined in Algorithm 2 in Appendix A. We learn our final hierarchical policy through $k$ stages of skill acquisition. Each of these stages is broken down into a base skill acquisition phase and a novel skill acquisition phase in a 2-phase curriculum learning.

In the base skill acquisition phase, we only sample tasks from the base task set $\mathcal{G}_{k-1}$. This ensures that the global policy learns how to use previously learned skills by issuing instructions to the base policy. In other words, this phase teaches the agent how to connect its instruction policy to its base policy. Once the average reward for all base tasks exceeds a certain threshold, we proceed to the next phase.

In the novel skill acquisition phase, we sample tasks from the full task set, $\mathcal{G}_k$, for the $k$-th stage of skill acquisition. It is in this phase that the agent can learn when to rely on the base policy and when to rely on the augmented flat policy for executing novel tasks.

In each of these phases, all policies are trained with advantage actor-critic (A2C) (Section 4.1) and distributions in the STG are estimated based on accumulated positive episodes (Section 4.2).

## 4.1 Policy Optimization by Advantage Actor-Critic

We use advantage actor-critic (A2C) for policy optimization with off-policy learning (Su et al., 2017). Here, we only consider the gradient for global policies (i.e., $k > 0$) as we assume the terminal policy has been trained as initial condition. Let $V_k(s_t, g)$ be a value function indicating the expected return given state $s_t$ and task $g$. To reflect the nature of the branch switching in our model, we introduce another value function $V_k^{\mathrm{sw}}(s_t, e_t, g)$ to represent the expected return given state $s_t$, task $g$ and current branch selection $e_t$.

Thus, given a trajectory $\Gamma = \{\langle s_t, e_t, g'_t, a_t, r_t, \mu_k^{\text{sw}}(\cdot|s_t), \mu_k^{\text{inst}}(\cdot|s_t, g), \mu_k^{\text{aug}}(\cdot|s_t, g), g \rangle : t = 0, 1, \cdots, T\}$ generated by old policies $\mu_k^{\text{sw}}(\cdot|s_t)$, $\mu_k^{\text{inst}}(\cdot|s_t, g)$, and $\mu_k^{\text{aug}}(\cdot|s_t, g)$, the policy gradient reweighted by importance sampling can be formulated as

$$
\underbrace{\omega_t^{\text{sw}} \nabla_{\theta^{\text{sw}}} \log \pi_k^{\text{sw}}(e_t|s_t, g) A(s_t, g, e_t)}_{\text{1st term: switch policy gradient}}
$$
$$
+ \underbrace{(1 - e_t)\omega_t^{\text{inst}} \nabla_{\theta^{\text{inst}}} \log \pi_k^{\text{inst}}(g'_t|s_t, g) A(s_t, g, e_t, g'_t)}_{\text{2nd term: instruction policy gradient}} \quad (8)
$$
$$
+ \underbrace{e_t \omega_t^{\text{aug}} \nabla_{\theta^{\text{aug}}} \log \pi_k^{\text{aug}}(a_t|s_t, g) A(s_t, g, e_t, a_t)}_{\text{3rd term: augmented policy gradient}},
$$

where $\omega_t^{\text{sw}} = \frac{\pi_k^{\text{sw}}(e_t|s_t, g)}{\mu_k^{\text{sw}}(e_t|s_t, g)}$, $\omega_t^{\text{inst}} = \frac{\pi_k^{\text{inst}}(g'_t|s_t, g)}{\mu_k^{\text{inst}}(g'_t|s_t, g)}$, and $\omega_t^{\text{aug}} = \frac{\pi_k^{\text{aug}}(a_t|s_t, g)}{\mu_k^{\text{aug}}(a_t|s_t, g)}$ are importance sampling weights for the three terms respectively; $A(s_t, g, e_t)$, $A(s_t, g, e_t, g'_t)$, and $A(s_t, g, e_t, a_t)$ are estimates of advantage functions, which have multiple possible definitions. In this paper, we define them by the difference between empirical return and value function estimation: $A(s_t, g, e_t) = \sum_{\tau=0}^{\infty} \gamma^\tau R(s_{t+\tau}, g) - V_k(s_t, g)$, $A(s_t, g, e_t, g'_t) = A(s_t, g, e_t, a_t) = \sum_{\tau=0}^{\infty} \gamma^\tau R(s_{t+\tau}, g) - V_k^{\text{sw}}(s_t, g, e_t)$, where $\gamma$ is the discounted coefficient.

Finally, the value functions can be updated using the following gradient:

$$
\nabla_{\theta_v} \frac{1}{2} \left[ \sum_{\tau=0}^{\infty} \gamma^\tau R(s_{t+\tau}, g) - V_k(s_t, g) \right]^2 + \nabla_{\theta_v^{\text{sw}}} \frac{1}{2} \left[ \sum_{\tau=0}^{\infty} \gamma^\tau R(s_{t+\tau}, g) - V_k^{\text{sw}}(s_t, e_t, g) \right]^2. \quad (9)
$$

To increase the episode efficiency, after running an episode, we conduct $n$ mini-batch updates where $n$ is sampled from a Poisson distribution with $\lambda = 4$, similar to Wang et al. (2017). Note that one can also apply other common policy optimization methods, e.g., A3C (Mnih et al., 2016), to our model. We leave this as future work to evaluate the efficiency of different methods when using our model.

Optimizing all three sub-policies together leads to unstable learning. To avoid this, we apply a simple alternating update procedure. For each set of $M$ iterations, we keep two of the sub-policies fixed and train only the single policy that remains. When we reach $M$ iterations, we switch the policy that is trained. For all experiments in this paper, we use $M = 500$. This alternating update procedure is used within both phases of curriculum learning.

## 4.2 LEARNING AN STG

If at any point in the aforementioned training process the agent receives a positive reward after an episode, we update the stochastic temporal grammar. $\rho_k$ and $q_k$ of the STG are both initialized to be uniform distributions. Since the STG is a finite state Markov chain over tuples $\langle e_t, g'_t \rangle$, we use maximum likelihood estimation (MLE) to update the distributions (Baum & Petrie, 1966). As the training progresses, the STG starts to guide the exploration.

To avoid falling into local minima in the early stages of training, it is important to encourage random exploration in early episodes. Based on our experiments, we find that using $\epsilon$-greedy suffices.

## 5 EXPERIMENTS

### 5.1 GAME ENVIRONMENT AND TASK SPECIFICATIONS

Figure 3 (left) shows the two room environment in Minecraft that we created using the Malmo platform (Johnson et al., 2016). In each episode, an arbitrary number of blocks with different colors (totaling 6 colors in our experiments) are randomly placed in one of the two rooms. The agent is initially placed in the same room with the items. We consider five sets of tasks: i) $\mathcal{G}^{(0)} = \{\text{"Find x"}\}$, walking to the front of a block with color x, ii) $\mathcal{G}^{(1)} = \{\text{"Get x"}\}$, picking up a block with color x, iii) $\mathcal{G}^{(2)} = \{\text{"Put x"}\}$, putting down a block with color x, iv) $\mathcal{G}^{(3)} = \{\text{"Stack x"}\}$, stacking two blocks with color x together, and v) $\mathcal{G}^{(4)} = \{\text{'Put x on y'}\}$, putting a block with color x on top of

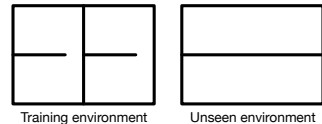

Figure 3: Room layout. Left: training environment; right: unseen rooms for testing.

a block with a different color y. In total, there are 54 tasks. An agent can perform the following actions: "move forward," "move backward," "move left," "move right," "turn left," "turn right," "pick up," "put down."

Without loss of generality, we assume the following skill acquisition order: $\mathcal{G}_k = \cup_{\kappa=1}^k \mathcal{G}^{(\kappa)}$, $\forall k = 0, 1, 2, 3, 4$, which is a natural way to increase skill sets. One may also alter the order, and the main conclusions shall still hold. This results in policies $\{\pi_k : k = 0, 1, 2, 3, 4\}$ for these four task sets. For the last task set, we hold out 6 tasks out of all 30 tasks (i.e., 3 pairs of colors out of 15 color combinations) for testing and the agent will not be trained on these 6 tasks.

We adopt a sparse reward function: when reaching the goal of a task, the agent gets a $+1$ reward; when generating an instruction $g'$ that is not executable in current game (e.g., trying to find an object that does not exist in the environment), we give a $-0.5$ reward; otherwise, no reward will be given. Whenever a non-zero reward is given, the game terminates. Note that the negative reward is only given during training.

## 5.2 IMPLEMENTATION DETAILS

We specify the architecture of the modules in our model in Appendix B, where the visual and instruction encoding modules have the same architectures as the ones in Hermann et al. (2017). We train the network with RMSProp (Tieleman & Hinton, 2012) with a learning rate of 0.0001. We set the batch size to be 36 and clip the gradient to a unit norm. For all tasks, the discounted coefficient is $\gamma = 0.95$. For the 2-phase curriculum learning, we set the average reward threshold to be 0.9 (average rewards are estimated from the most recent 200 episodes of each task).

To encourage random exploration, we apply $\epsilon$-greedy to the decision sampling for the global policy (i.e., only at the top level $k$ at each stage $k > 0$), where $\epsilon$ gradually decreases from 0.1 to 0.

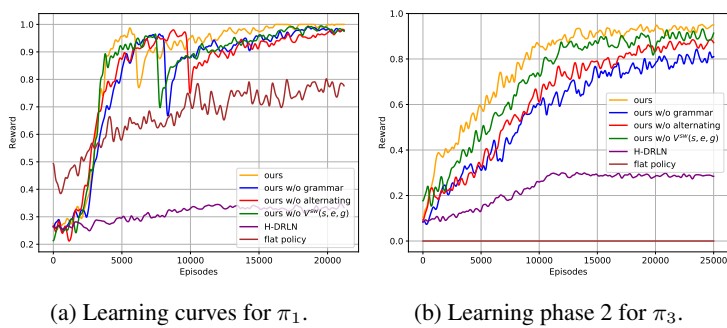

(a) Learning curves for $\pi_1$.  (b) Learning phase 2 for $\pi_3$.

Figure 4: Comparison of learning efficiency on two task sets: (a) $\mathcal{G}_1$ for global policy $\pi_1$ and (b) $\mathcal{G}_3$ for global policy $\pi_3$ respectively.

## 5.3 LEARNING EFFICIENCY

To evaluate the learning efficiency, we compare our full model with 1) a flat policy (Figure 2a) as in Hermann et al. (2017) fine-tuned on the terminal policy $\pi_0$, 2) H-DRLN (Tessler et al., 2017) and variants of our approach: 3) ours without STG, 4) ours without alternating policy optimization, and 5) ours without $V_k^{sw}(s, e, g)$ (replaced by $V_k(s, g)$ instead). Note that all the rewards have been converted to the same range, i.e., $[0, 1]$ for the sake of fair comparison.

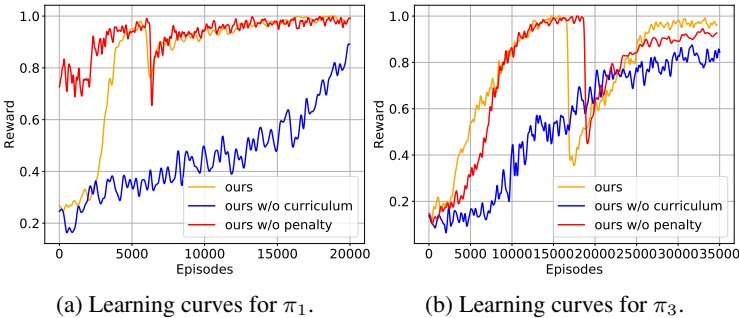

(a) Learning curves for $\pi_1$.   (b) Learning curves for $\pi_3$.

Figure 5: Effects of different training protocols.

Table 1: Success rates in different game settings including scenarios seen during training, new environments and new tasks. All policies are trained in the same environment in seen scenarios.

| Method | Seen scenarios | | | | | Unseen environments | | | | Unseen tasks |
|---|---|---|---|---|---|---|---|---|---|---|
| | Find x | Get x | Put x | Stack x | Put x on y | Find x | Get x | Put x | Stack x | Put x on y |
| Full model | 0.995 | 0.970 | 1.00 | 0.955 | 0.873 | 0.723 | 0.648 | 1.00 | 0.613 | 0.792 |
| Flat policy | 0.980 | 0.965 | 1.00 | 0 | 0 | 0.515 | 0.450 | 1.00 | 0 | 0 |

In Figure 4a, we use various methods to train policy $\pi_1$ for the task set $\mathcal{G}_1$ based on the same base policy $\pi_0$. The large dip in the reward indicates that the curriculum learning switches from phase 1 to phase 2. From Figure 4a, we may clearly see that our full model and variants can all converge within 22,000 episodes, whereas the average reward of the flat policy is still below 0.8 given the same amount of episodes. In addition, our full model finishes phase 1 significantly faster than other methods and its curve of average reward maintains notably higher than the remaining ones.

To further examine the learning efficiency during phase 2 when new tasks are added into the training process, we first pretrain $\pi_3$ using our full model following our definition of phase 1 in the curriculum learning. We then proceed to learning phase 2 using different approaches all based on this pretrained policy. As shown in Figure 4b, our full model has the fastest convergence and the highest average reward upon convergence. By comparing Figure 4a and Figure 4b, we further show that our full model has a bigger advantage when learning more complex tasks.

Since we have a large number of previously learned tasks, H-DRLN is clearly not able to learn a descent policy according the results. Note that an H-DRLN can only learn one task at a time, each of its curves in Figure 4 is for a single task (i.e., "Get white" and "Stack white" respectively).

To demonstrate the effects of our 2-phase curriculum learning and the $-0.5$ penalty on the training efficiency, we visualize the learning curves of our model trained without the curriculum learning or without the penalty along with the one trained with the full protocol in Figure 5. According to the results, the curriculum learning indeed helps accelerate the convergence, which empirically proves the importance of encouraging a global policy to reuse relevant skills learned by its base policy. It also appears that adding the penalty is an insignificant factor on learning efficiency except that it helps shorten the episode lengths as an episode ends whenever a penalty is given.

### 5.4 POLICY GENERALIZATION

Finally, we evaluate how the hierarchical design and encoding tasks by human instructions benefit the generalization of learned policies in the following three ways.

First, we train $\pi_1$ in a simpler setting where in each episode, only one item (i.e, the target item of the given task) is present. We then test the policy $\pi_1$ for "Get x" tasks in a room where there will be multiple items serving as distraction and the agent must interact with the correct one. Both the flat policy and the hierarchical policy can achieve near perfect testing success rate in the simple setting. However, in the more complex setting, flat policy can not differentiate the target item from other

items that are also placed in the room (the success rate drops to 29%), whereas our hierarchical policy still maintains a high success rate (94%). This finding suggests that the hierarchical policy not only picks up the concept of "find" and "get" skills as the flat policy does, but also inherits the concept of items from the base policy by learning to utter correct instructions to deploy "find" skill in the base policy.

Second, we reconfigure the room layout in Figure 3 (left) and test the flat policy and our full model in the new rooms shown in Figure 3 (right) for various tasks. Both policies are trained in the same environment. There are multiple items in a room for both training and testing cases. The success rates are summarized in Table 1. Using the flat policy results in a much bigger drop in the testing success rate compared to using out full model. This is mainly because that our global policy will repeatedly call its base policy to execute the same task until the agent finally achieves the goal even though the trained agent is unable to reach the goal by just one shot due to the simplicity of the training environment.

Third, we evaluate the learned policy on the 6 unseen tasks for the "Put x on y" tasks as a zero-short evaluation. The success rate reported in Table 1 suggests that our model is able to learn the decomposition of human instructions and generate correct hierarchical plans to perform unseen tasks.

## 5.5 Policy Interpretability

We visualize typical hierarchical plans of several tasks generated by global policies learned by our full model in Appendix C (Figure 6 and Figure 7)[1]. It can been seen from the examples that our global policies adjust the composed plans in different scenarios. For instance, in the second plan on the first row, $\pi_1$ did not deploy base policy $\pi_0$ as the agent was already in front of the target item at the beginning of the episode, whereas in the plan on the second row, $\pi_1$ deployed $\pi_0$ for the "Find x" base task twice consecutively, as it did not finish the base task in the first call.

## 6 Conclusion

In this work, we have proposed a hierarchal policy modulated by a stochastic temporal grammar as a novel framework for efficient multi-task reinforcement learning through multiple training stages. Each task in our settings is described by a human instruction. The resulting global policy is able to reuse previously learned skills for new tasks by generating corresponding human instructions to inform base policies to execute relevant base tasks. We evaluate this framework in Minecraft games and have shown that our full model i) has a significantly higher learning efficiency than a flat policy does, ii) generalizes well in unseen environments, and iii) is capable of composing hierarchical plans in an interpretable manner.

Currently, we rely on weak supervision from humans to define what skills to be learned in each training stage. In the future, we plan to automatically discover the optimal training procedures to increase the task set.

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

## A  Pseudo Code of Our Algorithms

---

**Algorithm 1** RUN($k, g$)

---

**Input:** Policy level $k$, task $g \in \mathcal{G}_k$
**Output:** Episode trajectory $\Gamma$ at the top level policy
1: $t \leftarrow 0$
2: $\Gamma = \emptyset$
3: Get initial state $s_0$
4: **repeat**
5:    **if** k == 1 **then**
6:       Sample $a_t \sim \pi_k(\cdot|s_t, g)$ and execute $a_t$
7:       Get current state $s_{t+1}$
8:       $r_t \leftarrow R(s_{t+1}, g)$
9:       Add $\langle s_t, a_t, r_t, \pi_k(\cdot|s_t, g), g\rangle$ to $\Gamma$
10:    **else**
11:       Sample $e_t$ and $g'_t$ as in Section 3.3 for using STG as guidance
12:       Sample $a_t \sim \pi_k^{\text{aug}}(\cdot|s_t, g)$
13:       **if** $e_t = 0$ **then**
14:          *// Execute base policy $\pi_{k-1}$ by giving instruction $g'_t$*
15:          RUN($k - 1, g'_t$)
16:       **else**
17:          Execute $a_t$
18:       **end if**
19:       Get current state $s_{t+1}$
20:       $r_t \leftarrow R(s_{t+1}, g)$
21:       Add $\langle s_t, e_t, g'_t, a_t, r_t, \pi_k^{\text{sw}}(\cdot|s_t), \pi_k^{\text{inst}}(\cdot|s_t, g), \pi_k^{\text{aug}}(\cdot|s_t, g), g\rangle$ to $\Gamma$
22:    **end if**
23:    $t \leftarrow t + 1$
24: **until** $t > T$ or $r_t \neq 0$

---

**Algorithm 2** Learning global policy and STG at stage $k > 0$

---

1: Specify $\lambda$, maximum training iterations $N$, alternating update rotation frequency $M$, and reward threshold $R_{\min}$
2: Initialize total replay memory $D \leftarrow \emptyset$ and its subset for positive episodes $D_+ \leftarrow \emptyset$
3: Initialize current iteration id $i \leftarrow 0$ and set current updating term to be $\tau \leftarrow 1$
4: Initialize parameters of policies and value functions $\Theta = \langle \theta^{\text{sw}}, \theta^{\text{inst}}, \theta^{\text{aug}}, \theta_v, \theta_v^{\text{sw}}\rangle$
5: Initialize distributions of the STG as uniform distributions
6: **repeat**
7:    Determine current learning phase by comparing average rewards of tasks in $\mathcal{G}_{k-1}$ with $R_{\min}$
8:    **if** in curriculum learning phase 1 **then**
9:       Sample a task $g$ from base task set $\mathcal{G}_{k-1}$
10:    **else**
11:       Sample a task $g$ from global task set $\mathcal{G}_k$
12:    **end if**
13:    //Run an episode
14:    $\Gamma \leftarrow$ RUN($k, g$)
15:    $D \leftarrow D \cup \Gamma$
16:    **if** the maximum reward in $\Gamma$ is +1 **then**
17:       $D_+ \leftarrow D_+ \cup \Gamma$
18:       Re-estimate the distributions of the STG based on updated $D_+$ by MLE
19:    **end if**
20:    Sample $n \sim \text{Possion}(\lambda)$
21:    **for** $j \in \{1, \cdots, n\}$ **do**
22:       Sample a mini-batch $S$ from $D$
23:       Update $\Theta$ based on (9) and the $\tau$-th term in (8)
24:       $i \leftarrow i + 1$
25:       **if** $i\%M = 0$ **then**
26:          $\tau \leftarrow \tau\%3 + 1$
27:       **end if**
28:    **end for**
29: **until** $i \geq N$

---

# B ARCHITECTURES OF MODULES

The architecture designs of all modules in our model shown in Figure 2 are as follows:

**Visual Encoder** extracts feature maps from an input RGB frame with the size of $84 \times 84$ through three convolutional layers: i) the first layer has 32 filters with kernel size of $8 \times 8$ and stride of $4$; ii) the second layer has 64 filters with kernel size of $4 \times 4$ and stride of $2$; iii) the last layer includes 64 filters with kernel size of $3 \times 3$ and stride of $1$. The feature maps are flatten into a 3136-dim vector. We reduce the dimension of this vector to 256 by a fully connected (FC) layer resulting a 256-dim visual feature as the final output of this module.

**Instruction Encoder** first embeds each word into a 128-dim vector and combines them into a single vector by bag-of-words (BOW). Thus the output of this module is a 128-dim vector. For more complex instructions such as "Put x on y", we replace BOW by a GRU with 128 hidden units.

**Fusion** layer simply concatenates the encoded visual and language representations together and outputs 384-dim fused representation. We then feed this 384-dim vector into an **LSTM** with 256 hidden units. The hidden layer output of the LSTM is served as the input of all policy modules and value function modules.

**Switch Policy** module has a FC layer with output dimension of 2 and a softmax activation to get $\pi_k^{\text{sw}}(e|s,g)$. **Instruction Policy** module has two separate FC layers, both of which are activated by softmax to output the distribution of *skill*, $p_k^{\text{skill}}(u_{\text{skill}}|s,g)$, and the distribution of *item*, $p_k^{\text{item}}(u_{\text{item}}|s,g)$, respectively. **Augmented Policy** module outputs $\pi_{\text{aug}}(a|s,g)$ also through a FC layer and softmax activation. The two **Value Function** modules, $V(s,g)$ and $V^{\text{sw}}(s,e,g)$, all have a scalar output through a FC layer.

Finally, the **Selector** module selects the action sampled from **Augmented Policy** module or **Base Policy** module based on the switching decision sampled from the Switch Policy module.

# C COMPOSED HIERARCHICAL PLANS

Figure 6 and Figure 7 show several plans for different tasks composed by executing our hierarchical policies.

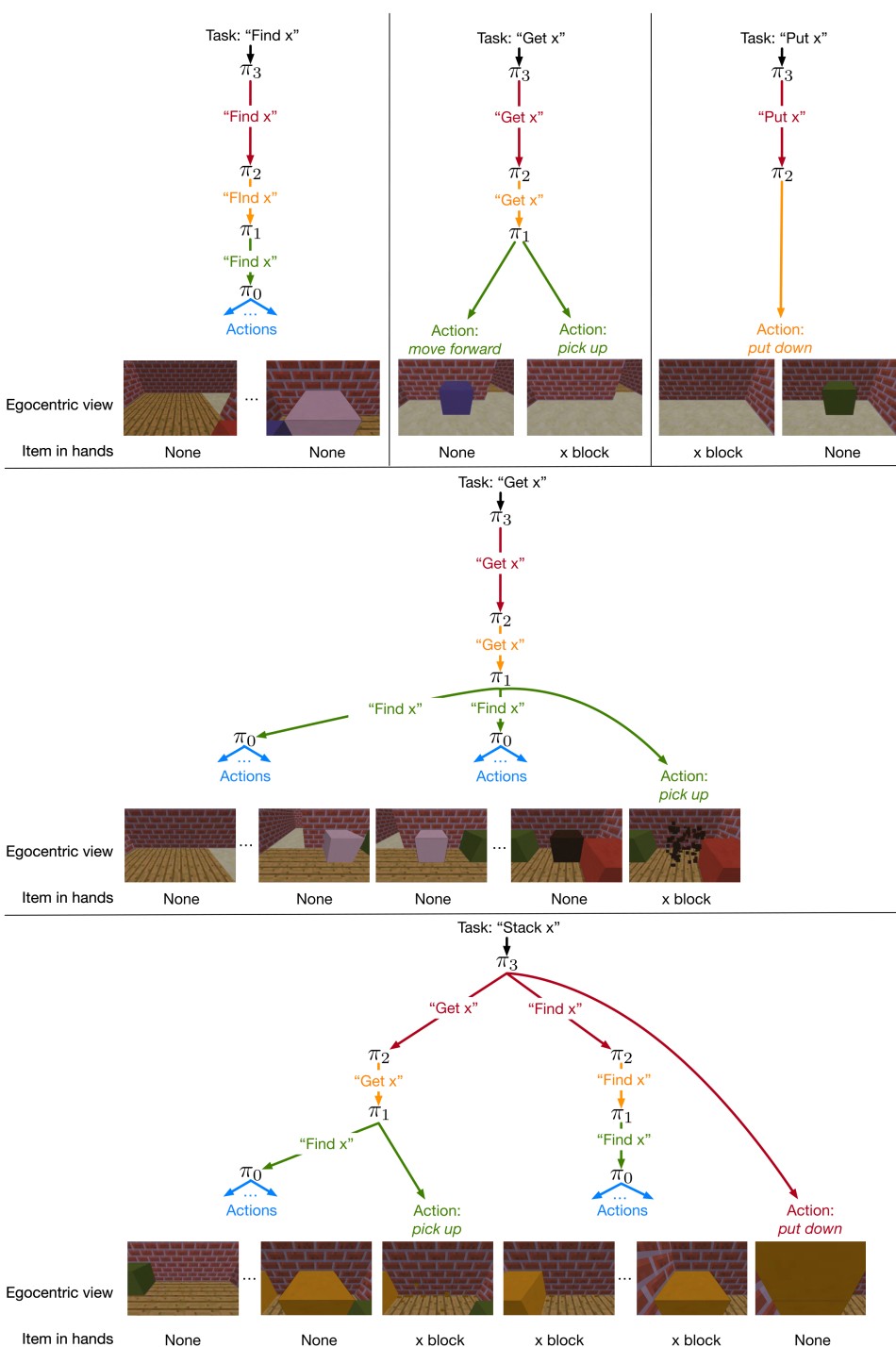

Figure 6: Samples of typical hierarchical plans for different tasks composed by our global policies. Note that all tasks must start from the top-level policy. The branches are ordered from left to right in time indicating consecutive steps carried out by a policy. We also show the egocentric view and the item in hands at critical moments for a real episode example.

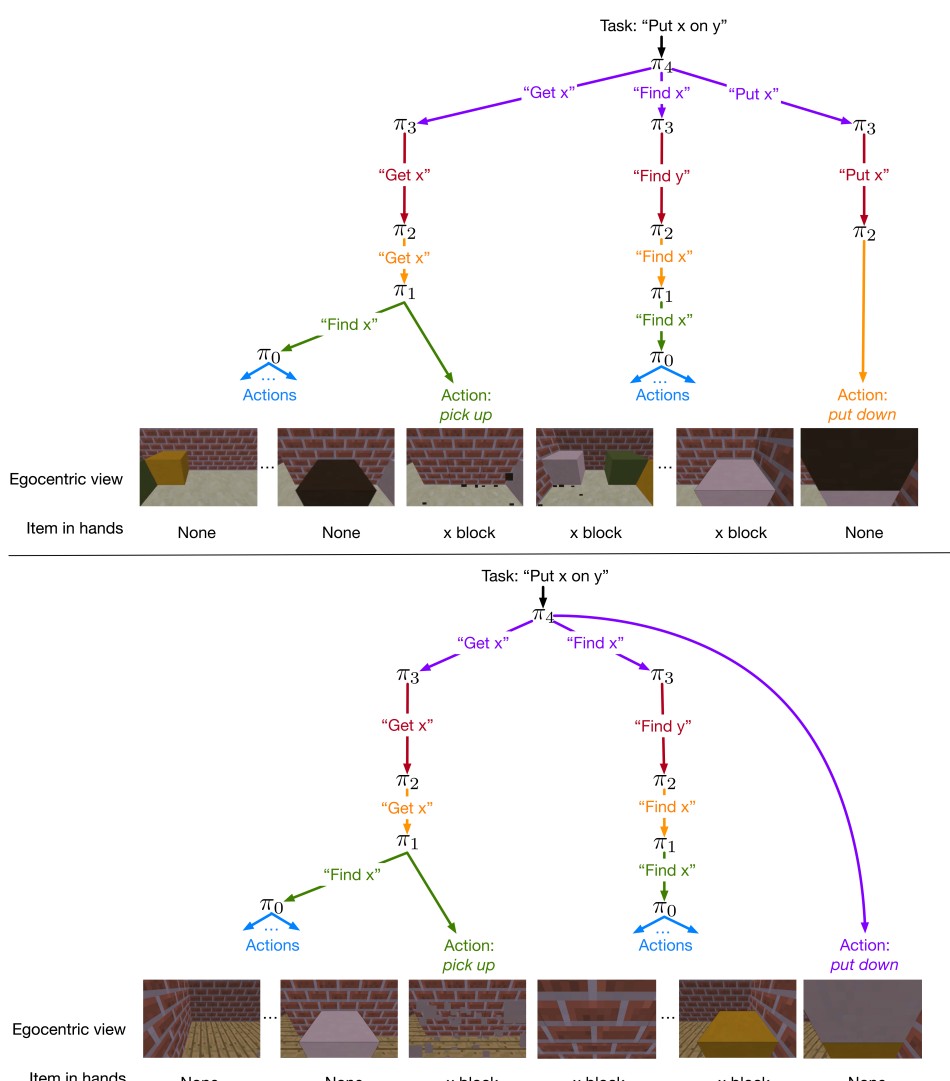

Figure 7: Hierarchical plans for "Put x on y" tasks. Top: an example of performing trained tasks; bottom: an example of generalizing the plan composition to unseen tasks.

