# OpenReview forum: "Hierarchical and Interpretable Skill Acquisition in Multi-task Reinforcement Learning"
_ICLR.cc/2018/Conference — Accept (Poster)_

### Official Review · AnonReviewer3 · 2017-11-25
**A good idea, but with limited evaluation**

**Rating:** 6
**Confidence:** 4

**Review:**

This paper aims to learn hierarchical policies by using a recursive policy structure regulated by a stochastic temporal grammar. The experiments show that the method is better than a flat policy for learning a simple set of block-related skills in minecraft (find, get, put, stack) and generalizes better to a modification of the environment (size of room). The sequence of subtasks generated by the policy are interpretable.

Strengths:
- The grammar and policies are trained using a sparse reward upon task completion.
- The method is well ablated; Figures 4 and 5 answered most questions I had while reading.
- Theoretically, the method makes few assumptions about the environment and the relationships between tasks.
- The interpretability of the final behaviors is a good result.

Weaknesses:
- The implementation gives the agent a -0.5 reward if it generates a currently unexecutable goal g’. Providing this reward requires knowing the full state of the world. If this hack is required, then this method would not be useful in a real world setting, defeating the purpose of the sparse reward mentioned above. I would really like to see how the method performs without this hack.
- There are no comparisons to other multitask or hierarchical methods. Progressive Networks or Zero-Shot Task Generalization with Multi-Task Deep Reinforcement Learning seem like natural comparisons.
- A video to show what the environments and tasks look like during execution would be helpful.
- The performances of the different ablations are rather close. Please a standard deviation over multiple training runs. Also, why does figure 4.b not include a flat policy?
- The stages are ordered in a semantically meaningful order (find is the first stage), but the authors claim that the order is arbitrary. If this claim is going to be included in the paper, it needs to be proven (results shown for random orderings) because right now I do not believe it.

Quality:
The method does provide hierarchical and interpretable policies for executing instructions, this is a meaningful direction to work on.

Clarity:
Although the method is complicated, the paper was understandable.

Originality and significance:
Although the method is interesting, I am worried that the environment has been too tailored for the method, and that it would fail in realistic scenarios. The results would be more significant if the tasks had an additional degree of complexity, e.g. “put blue block next to the green block” “get the blue block in room 2”. Then the sequences of subtasks would be a bit less linear (e.g., first need to find blue, then get, then find green, then put). At the moment the tasks are barely more than the actions provided in the environment.

Another impedance to the paper’s significance is the number of hacks to make the method work (ordering of stages, alternating policy optimization, first training each stage on only tasks of previous stage). Because the method is only evaluated on one simple environment, it unclear which hacks are for the method generally, and which hacks are for the method to work on the environment.

---

> ### Author Response · Authors · 2018-01-04
> **Author response**
>
> Thank you for your reviews and insights.
>
> 1. “Providing this reward requires knowing the full state of the world”
>
> Learning curves of training without the penalty have been added to Figure 5. We find that the penalty does not have a significantly effect on the learning efficiency as shown in Figure 5.
>
> Reviewer may be confused by our phrasing. Sorry about that. Actually we want to stress that we do not provide the full state when using this penalty. Instead, a penalty only includes information about the agent’s physical capacity and the nature of the target task, since it is given i) when the agent attempts to execute tasks that exceeds its physical capacity, such as trying to put down a block when it is not carrying one or trying to pick-up another block where there is already one in its hands, ii) or when it attempts to execute tasks that are irrelevant to that tasks.
>
> Also, we only use this during training under the assumption that a given task in the training is always executable (the necessary blocks are present in the environment). So when a penalty was given to a task of finding an object that does not exist in the environment, it was meant to save time in game playing and also gives training signical of what old tasks are relevant for the agent.
>
> Finally, in testing, we do not use this penalty, and it does not affect the performance.
>
> Therefore, we do not think that this will prevent us to apply the approach to the real world for the aforementioned reasons.
>
> 2. “There are no comparisons to other multitask or hierarchical methods”
> We have also evaluated H-DRLN (Tessler et al., 2017).
>
> 3. “A video … would be helpful”
> Please refer to this link for the video (with audio): https://www.dropbox.com/s/j5nw2cljpoofo9j/hrl_demo.mov?dl=0
>
> 4. “The performances of the different ablations are rather close. Please a standard deviation over multiple training runs.”
> We do not include the standard deviations since there are no noticeable difference among them. The alternating and 2 value functions mainly help accelerating the learning phase 1. As shown in Figure 4a, the acceleration is significant (full model is the first one that switches to phase 2). In phase 2 (see Figure 4b), the advantage of using the STG is very clear (the blue curve reaches a plateau around an average reward of 0.8) and the others do not show a large improvement over the full model. So none of them reduces the training variance, but they all help increase the training efficiency.
>
> 5. “why does figure 4.b not include a flat policy””
> We have added the flat policy.
>
> 6. “the authors claim that the order is arbitrary”
> Sorry for the confusion. We will clarify this. What we meant was for arbitrary order, we can still train the hierarchical policy, but a semantically meaningful order is indeed very important for the training efficiency. And we do provide this order as weak supervision.
>
> 7. “unclear which hacks are for the method generally, and which hacks are for the method to work on the environment.”
> They are not environment dependent.
> i) The order of the stages comes from semantic meanings of the tasks, so it depends on the tasks but does not depend on the environment. E.g., you may train a real robot on the same tasks in the same order in a real environment. In fact, the generated interpretable hierarchical plans can be directly used for the same tasks in different environments without additional training as long as there are equivalent primitive actions.
> ii) Alternating is purely for the optimization of the neural nets. As the experimental results show, we can also train the hierarchical policies without it.
> iii) The first phase in the 2-phase curriculum is for the global policy to learn what the goals of the previous tasks are, so that in the second phase, it knows when to repeat the same task and when to stop. This is also not tailored to any specific environment.

---

> ### Comment · AnonReviewer3 · 2018-01-13
> **Review revision**
>
> I am happy with the updates the authors made to the paper. The video and additional experiments are valuable, and I will increase my score accordingly.
>
> However, the paper would strongly benefit from either a second test environment or a more complex grammar to showcase generality.
>
> (nitpick): “We do not include the standard deviations since there are no noticeable difference among them.” including standard deviations is not just to compare the magnitude of the standard deviations, but to see whether the differences in means of the methods are within the standard deviations.

---

### Official Review · AnonReviewer1 · 2017-11-27
**Hierarchical and Interpretable Skill Acquisition in Multi-Task Reinforcement Learning**

**Rating:** 6
**Confidence:** 3

**Review:**

Summary:
This paper proposes an approach to learning hierarchical policies in a lifelong learning context. This is achieved by stacking policies - an explicit "switch" policy is then used to decide whether to execute a primitive action or call the policy of the layer below it. Additionally, each task is encoded in a human-readable template, which provides interpretability.
Review:
Overall, I found the paper to be generally well-written and the core idea to be interesting. My main concern is about the performance against existing methods (no empirical results are provided), and while it does provide interpretability, I am not sure that other approaches (e.g. Tessler et al. 2017) could not be slightly modified to do the same. I think the paper could also benefit from at least one more experiment in a different, harder domain.

I have a few questions and comments about the paper:

The first paragraph claims "This precludes transfer of previously learned simple skills to a new policy defined over a space with differing states or actions". I do not see how this approach avoids suffering from the same problem? Additionally, approaches such as agent-space options [Konidaris and Barto. Building Portable Options: Skill Transfer in Reinforcement Learning, IJCAI 2007] get around at least the state part.

I do not quite follow what is meant by "a global policy is assumed to be executable by only using local policies over specific options". It sounds like this is saying that the inter-option policy can pick only options, and not primitive actions, which is obviously untrue. Can you clarify this sentence?

In section 3.1, it may be best to mention that the policy accepts both a state and task and outputs an action. This is stated shortly afterwards, but it was confusing because section 3.1 says that there is a single policy for a set of tasks, and so obviously a normal state-action policy would not work here.

At the bottom of page 6, are there any drawbacks to the instruction policy being defined as two independent distributions? What if not all skills are applicable to all items?

In section 5, what does the "without grammar" agent entail? How is the sampling from the switch and instruction policies done in this case?

While the results in Figures 4 and 5 show improvement over a flat policy, as well as the value of using the grammar, I am *very* surprised there is no comparison to existing methods. For example, Tessler's H-DRLN seems like one obvious comparison here, since it learns when to execute a primitive action and when to reuse a skill.

There were also some typos/small issues (I may have missed some):

pg 3: "In addition, previous work usually useS..."
pg 3. "we encode a human instruction to LEARN A..." (?)
pg 4. "...with A stochastic temporal grammar..."
pg 4. "... described above through A/THE modified..."
pg 6. "...TOTALLING six colors..."
There are some issues with the references (capital letters missing e.g. Minecraft)

It also would be preferable if the figures could appear after they are referenced in the text, since it is quite confusing otherwise. For example, Figure 2 contains V(s,g), but that is only defined much later on. Also, I struggled to make out the yellow box in Figure 2, and the positioning of Figure 3 on the side is not ideal either.

---

> ### Author Response · Authors · 2018-01-04
> **Author response**
>
> Thank you for your comments and suggestions.
>
> 1. “performance against existing methods”
> We have added the comparison with Tessler et al. (2017), i.e., H-DRLN, as suggested (see Table 1 and Figure 4). H-DRLN indeed also has the concept of reusing previously learned skills but the advantages of our approach are very obvious:
> 1) Each H-DRLN in Tessler et al. (2017) can only learn one task like “Stack blue,” whereas ours can learn a set of tasks, like {“Stack x”}, where x can be different colors. In fact, each curve of H-DRLN we show in Figure 4 is only for training one particular task (i.e., “Get white” and “Stack white” respectively), whereas all the other curves are for training the whole set of tasks.
> 2) H-DRLN treats all the old tasks {DSN_i} and the actions on the same level and learns Q(s, a) and Q(s, DSN_i). This setting is clearly not scalable. In the original paper, the learning was done where there were only 4 old tasks. However, in our settings, we have as many as 18 old tasks. And clearly from the results, H-DRLN can not learn good policies for new tasks as the input space of the Q(s, DSN_i) function becomes unbearably large (it is similar to learning a policy based on a very large action space).
> 3) Our policy is also learning the semantics of the tasks from the instructions, thus it has better generalization. As we show in the experiments on the {“Put x on top of y”} tasks, ours generalizes well in noval (x, y) combinations unseen in training whereas H-DRLN does not have this capability.
>
> 2. “one more experiment in a different, harder domain”
> We have tested more difficult tasks {“Put x on top of y”}, and have also evaluated our hierarchical policy in a zero-shot setting for unseen (x, y) combinations. See Table 1, Figure 4 and also Section 5.3 and Section 5.4. As shown in the results, flat policy and H-DRLN (Tessler et al., 2017) fail to learn good policies for {”Stack x”} tasks and the new {“Put x on top of y”} tasks. This manifests that the tasks are actually quite challenging for current RL methods.
>
> 3. Questions about the paper:
>
> 1) “I do not see how this approach avoids suffering from the same problem”
> Policies on different levels are learned on different corpus (e.g., pi_0 does not know the word “get”) and on different action spaces (e.g., pi_0 can be trained on actions excluding “pick up” and “put down”).  They also do not need to share a same state encoding module, so the forms of states can be different (e.g., we may use symbolic states for a global policy while its local policy uses raw pixels as states).
>
> 2) “It sounds like this is saying that the inter-option policy can pick only options, and not primitive actions”
> We are referring to some of the existing methods like Kulkarni et al. (2016), Andreas et al. (2017) where the set of necessary options for a global policy is predefined and a task is executed only based on these given options. Other methods like Tessler et al. (2017) do not have this limitation. We will clarify this.
>
> 3) “are there any drawbacks to the instruction policy being defined as two independent distributions”
> It is for simplicity. For more complex instructions, we may use GRUs or LSTMs to train the instruction generator, but the training will be slower.
>
> 4) “How is the sampling from the switch and instruction policies done in this case?”
> As we stated in the text, the sampling was done w.r.t. equation (1) and (2) when not using the STG.
>
> We agree with the other editing comments and have fixed them in the updated version.

---

### Official Review · AnonReviewer2 · 2017-11-27
**Interesting model but not very clear explanation, a bit weak experimental section.**

**Rating:** 6
**Confidence:** 3

**Review:**

This paper introduces an iterative method to build hierarchical policies. At every iteration, a new meta policy feeds in task id to the previous policy and mixes the results with an 'augmented' policy. The resulting policy is somewhat interpretable as the task id being sampled by the meta policy corresponds to one of the subgoals that are manually designed.

One of the limitation of the method is that appropriate subgoals and curriculum must be hand designed. Another one is that the model complexity grows linearly with the number of meta iterations.

The comparison to non-hierarchical models is not totally fair in my opinion. According to the experiment, the flat policy performs much worse than the hierarchical, but it is unclear how much of this is due to the extra capacity of the model of the unfolded hierarchical policy and how much of that is due to the hierarchy. In other words, it is unclear if hierarchy is actually useful, or just the task curriculum and model capacity staging.

The paper does not appear to be  fully self contained in term of notations, in particular regarding the importance sampling I could not find the definitions of mu, and regarding the STG I could not find the definition of q and rho.

The experimental results are a bit confusing. In the learning curves that are shown, it is not clear exactly when the set of task is expanded, nor when the hierarchical policy iteration occurs. Also, some curves are lacking the flat baseline.

---

> ### Author Response · Authors · 2018-01-04
> **Author response**
>
> Thank you for your reviews.
>
> 1. “a new meta policy feeds in task id to the previous policy”
> Actually, the global policy feeds an instruction in human language rather than a task ID to the previous policy. Compared to using task IDs, this i) improves the interpretability of the policy and ii) facilitates the generalization of the policy in noval scenarios/tasks thanks to the semantics of the instructions.
>
> 2. “appropriate subgoals and curriculum must be hand designed”
> Actually we let the global policy at each level to explore the appropriate subgoals for the new tasks among all previously learned tasks. The learning efficiency of our approach does depend on the given curriculum as existing curriculum-based RL training approaches do. We regard this as a weak supervision from human knowledge. In fact, compared to some recent work (Kulkarni  et al., 2016; Andreas et al., 2017) which specifically provide the necessary sub-goals and/or the order of the subgoals for a task, we do not think our setting is any less general.
>
> 3. “The comparison to non-hierarchical models is not totally fair”
> First, the training of flat policy is also strictly following the curriculum we use for our model. It is always finetuned based on the policy trained for the previous task set.  Second, the biggest benefit of our hierarchical policy comes from the more efficient exploration thanks to reusing old skills and learning the STG. As the updated Figure 4b shows, when training for more complex tasks, the flat policy can not yield any positive reward as achieving the goals requires a fairly long sequence of primitive actions and precise operations (e.g., picking up and putting down the correct blocks at the correct locations).
>
> 4. “I could not find the definitions of mu, and regarding the STG I could not find the definition of q and rho”
> Mu was introduced in the second paragraph of Section 4.1. q and rho were defined in the second paragraph of Section 3.3.
>
> 5. “The experimental results are a bit confusing”
> Sorry for the confusion. As we explained in Section 4, we adopt a 2-phase curriculum learning where the task set was expanded in the second phase. For curriculum-based training, the dip of a curve comes from the switch from phase 1 to phase 2, thus also indicates when the task set was expanded. For non-curriculum based training where there is no dip in the reward, the expansion starts from the first episode. Note that in Figure 4b, we only show the phase 2 of our curriculum, so the expansion starts from the first episode for all curves in this figure.
>
> We have also added the flat baseline for the Figure 4b. The reason we didn’t include that in the previous version was that the flat policy failed to learn anything meaningful for the complex tasks.

---

### Author Response · Authors · 2018-01-04
**Submission revision**

Our updated submission has included extensive experiments as suggested by the reviewers. We have added a set of more challenging tasks with a zero-shot evaluation and also results of new baselines including an existing multi-task hierarchical policy method, H-DRLN in Tessler et al. (2017). The updated results can be seen in Section 5. A demo video (audio included) is available here: https://www.dropbox.com/s/j5nw2cljpoofo9j/hrl_demo.mov?dl=0

---

### Decision · Program_Chairs · 2018-01-29
**ICLR 2018 Conference Acceptance Decision**

**Decision:**

Accept (Poster)

**Comment:**

This method has a lot of strong points, but the reviewers had concerns about baselines, comparisons, and hand-engineered aspects of the method. The authors gave a strong rebuttal and made substantial updates to the paper to address the concerns. I think that this has saved the submission and tipped the balance towards acceptance.